# Electrocatalytic Degradation of Levofloxacin, a Typical Antibiotic in Hospital Wastewater

**DOI:** 10.3390/ma14226814

**Published:** 2021-11-11

**Authors:** Hongxia Lv, Peiwei Han, Xiaogang Li, Zhao Mu, Yuan Zuo, Xu Wang, Yannan Tan, Guangxiang He, Haibo Jin, Chenglin Sun, Huangzhao Wei, Lei Ma

**Affiliations:** 1Beijing Key Laboratory of Fuels Cleaning and Advanced Catalytic Emission Reduction Technology, College of New Materials and Chemical Engineering, Beijing Institute of Petrochemical Technology, Beijing 102617, China; 2019520036@bipt.edu.cn (H.L.); micklxg@163.com (X.L.); 2018310072@bipt.edu.cn (Y.Z.); wxonly0120@163.com (X.W.); hgx@bipt.edu.cn (G.H.); jinhaibo@bipt.edu.cn (H.J.); 2Guangzhou Institute of Energy Conversion, Chinese Academy of Sciences, School of Energy Science and Engineering, University of Science and Technology of China, Guangzhou 510640, China; hanpeiwei025@163.com; 3Institute of Applied Chemical Technology for Oilfield, College of New Materials and Chemical Engineering, Beijing Institute of Petrochemical Technology, Beijing 102617, China; muzhao@bipt.edu.cn; 4Dalian Institute of Chemical Physics, Chinese Academy of Sciences, Dalian 116023, China; yntan@dicp.ac.cn (Y.T.); clsun@dicp.ac.cn (C.S.)

**Keywords:** titanium suboxide electrode, levofloxacin, response surface methodology (RSM), degradation mechanisms and pathways

## Abstract

Presently, in the context of the novel coronavirus pneumonia epidemic, several antibiotics are overused in hospitals, causing heavy pressure on the hospital’s wastewater treatment process. Therefore, developing stable, safe, and efficient hospital wastewater treatment equipment is crucial. Herein, a bench-scale electrooxidation equipment for hospital wastewater was used to evaluate the removal effect of the main antibiotic levofloxacin (LVX) in hospital wastewater using response surface methodology (RSM). During the degradation process, the influence of the following five factors on total organic carbon (TOC) removal was discussed and the best reaction condition was obtained: current density, initial pH, flow rate, chloride ion concentration, and reaction time of 39.6 A/m^2^, 6.5, 50 mL/min, 4‰, and 120 min, respectively. The TOC removal could reach 41% after a reaction time of 120 min, which was consistent with the result predicted by the response surface (40.48%). Moreover, the morphology and properties of the electrode were analyzed. The degradation pathway of LVX was analyzed using high-performance liquid chromatography–mass spectrometry (LC–MS). Subsequently, the bench-scale electrooxidation equipment was changed into onboard-scale electrooxidation equipment, and the onboard-scale equipment was promoted to several hospitals in Dalian.

## 1. Introduction

With the continuous improvement of the human quality of life and the abuse of antibiotics, the application of antibiotics in various fields, such as medical care, agriculture, animal husbandry, aquaculture, and other fields [1], has become increasingly extensive. Fluoroquinolones (FQs) are broad-spectrum antibiotics with the largest consumption [2] and which are highly concentrated in the environment [3]. FQs primarily include levofloxacin (LVX), ofloxacin, norfloxacin, enrofloxacin, and ciprofloxacin [4]. Among them, LVX is the most widely used, mainly for treating pneumonia, urinary tract infection, acute pyelonephritis, and skin and tissue infections [5]. Actual novel coronavirus wastewater was tested, and the results are shown in Figure 1. The LVX content was the highest among all antibiotics. Therefore, LVX was selected as the target pollutant herein. The long-term existence and accumulation of LVX in water bodies is hazardous to human health, the natural environment, and ecological balance. Therefore, an effective degradation method is urgently needed to degrade LVX.

Existing methods for degrading antibiotics mainly include the ozone oxidation method [6], Fenton oxidation method [7,8], catalytic wet oxidation method [9,10], photocatalytic oxidation method [11,12], and electrocatalytic oxidation [13,14] methods. Among them, the electrocatalytic oxidation method has received wide attention from scholars because of its excellent selectivity, high degradation efficiency, few byproducts, and operability under normal temperature and pressure [15,16,17,18]. Presently, the commonly used anode materials in electrocatalysis mainly include boron-doped diamond (BDD) film [19,20,21,22,23], titanium suboxide [15,24,25,26,27], and carbon material [28,29,30] electrodes and dimensionally stable anodes (DSAs) electrodes [31,32,33,34,35,36]. In terms of the physical properties of titanium suboxide electrodes, the Magneli phase titanium oxide material (Ti_n_O_2n−1_) has good electrical conductivity under normal temperature [37]. Additionally, Magnéli phase titanium oxide materials, compared to the conventional electrode materials used in industry, exhibit high chemical stability, corrosion resistance, and a wide electrochemical stability potential window [15,26,37]. The material could be used as both cathode and anode, which is eco-friendly and cost-effective [38]. Therefore, herein, it was used as an electrooxidation anode, and ruthenium–titanium was used as a cathode to degrade target pollutants. As far as we know, this approach is the first time that a titanium suboxide electrode is being used to degrade LVX wastewater.

A single factor could affect the total organic carbon (TOC) removal, but note that interaction between factors could also affect the TOC removal. Response surface methodology (RSM) could explore the interaction among different influencing factors and fit the best conditions through multiple quadratic regression equations [39,40,41]. Thus, RSM was used to conduct electrooxidation and degradation of LVX model wastewater. In the degradation process, the influence of five factors, namely current density, initial pH value, flow rate, chloride ion concentration, and reaction time, were discussed.

Presently, in the context of the novel coronavirus pneumonia epidemic, several antibiotics are overused in hospitals, resulting in a large amount of medical wastewater containing antibiotics and the novel coronavirus. Since the antibiotics and novel coronavirus are difficult to degrade using traditional biochemical methods, it causes heavy pressure on the hospital wastewater treatment process. Therefore, it is necessary to develop stable, safe, and efficient hospital wastewater treatment equipment. Herein, brand-new hospital wastewater bench-scale and onboard-scale electrooxidation treatment equipment was developed. The bench-scale electrooxidation equipment for hospital wastewater was used to evaluate the removal effect of the main antibiotic LVX in hospital wastewater, which proved the stability and high efficiency of the equipment. Afterward, the bench-scale electrooxidation equipment was changed into onboard-scale electrooxidation equipment, and its application was promoted to many hospitals in Dalian.

The electrode was fully characterized by investigating the morphology and elemental composition of the material using scanning electron microscopy (SEM), X-ray diffraction (XRD), atomic force microscopy (AFM), and energy-dispersive X-ray spectrometry (EDS). A mathematical model of the influence of reaction conditions on the TOC removal rate was established using response surface methodology. The degradation pathway of LVX was analyzed using high-performance liquid chromatography–mass spectrometry (LC–MS). Subsequently, the bench-scale electrooxidation equipment was changed into onboard-scale electrooxidation equipment and promoted to many hospitals in Dalian.

## 2. Experimental Section

### 2.1. Experimental Materials

Levofloxacin (Shanghai Aladdin Biochemical Technology Co., Ltd., Shanghai, China, HPLC grade) and concentrated sulfuric acid (Liaoning Xinxing Reagent Company, Ltd., Tieling, China, 98%). Sodium hydroxide (Tianjin Kermel Chemical Reagent Co., Ltd., Tianjin, China, chemical purity), sodium sulfate (Meryer, Shanghai, China) Chemical Technology Co., Ltd., Shanghai, China, analytical purity), and sodium chloride (Beijing Chemical Plant, Beijing, China, analytical purity). Ultrapure water was used as the laboratory water.

### 2.2. Experimental Instrument

Total organic carbon analyzer (TOC-L, Shimadzu (Suzhou) Instruments Manufacturing Co., Ltd. Suzhou, China); DC stabilized power supply (HY3005MT, Hangzhou Huayi Electronics Industry Co., Ltd., Hangzhou, China); ultrasonic cleaning machine (SB-25-12DT, Ningbo Xinzhi Biological Technology Co., Ltd., Ningbo, China); tube furnace (SK2-4-12, Tianjin Zhonghuan Experimental Electric Furnace Co., Ltd., Tianjin, China); syringe filter (25-mm diameter, 0.45-μm pore size, membrane material of polyethersulfone (PES), Tianjin Jinteng Experimental Equipment Co., Ltd., Tianjin, China); fluorescence spectrophotometer (F4700, Hitachi, Tokyo, Japan).

### 2.3. Electrode Preparation

The ruthenium–titanium electrode was purchased from Baoji Eike Metal. The specific process of preparing the titanium suboxide electrode is as follows. The titanium plate was cut using a computer numerical control machine into a circle with a diameter of 80 mm, cleaned using absolute ethanol, and polished using a semi-automatic metallographic grinding and polishing machine. Before the deposition of the TiO_2_ coating, the surface of the titanium plate had been etched with oxygen plasma for 3 min at 100 W of RF power. Subsequently, we employed plasma-enhanced chemical vapor to deposit TiO_2_ onto the titanium plate at a power level of 200 W, flow rate of oxygen = 40 mL/min, flow rate of argon carrier gas = 2 mL/min, and the temperature of the TiCl_4_ container was kept at 0 °C. The system pressure during the deposition was ~53.2 Pa. The deposition process lasted 45 min. We were able to get a Ti_n_O_2n−1_-coated titanium plate. The titanium dioxide plate was reduced in a mixture of N_2_ and H_2_ gases, where the mixed gas was obtained instantaneously from decomposed ammonia. Ammonia flowed into an ammonia-decomposing furnace, and the decomposed gas directly went into the pipe reduction chamber, where the reduction of TiO_2_ occurred at 1127 K. The flow rate of ammonia was 1 L/min. The effective diameter of the electrodes was 75 mm [42,43].

### 2.4. TOC and LVX Content Analysis

The TOC of the samples was acquired using a TOC Analyzer (TOC-L CPN, Shimadzu, Tyoto, Japan). TOC removal efficiency was calculated as follows (Equation (1)):(1)$$\mathrm{TOC}\;\mathrm{removal}=\frac{\left({\mathrm{TOC}}_{0}-{\mathrm{TOC}}_{t}\right)}{{\mathrm{TOC}}_{0}}\times 100\%$$
where TOC_0_ is the total organic carbon of the initial wastewater, and TOC_t_ is the total organic carbon of the wastewater at the given time.
(2)$$\mathrm{Levofloxacin}\;\mathrm{Conversion}\;\mathrm{rates}=\frac{\left(C_{0}-C_{t}\right)}{C_{0}}\times 100\%,$$
where C_0_ is LVX content of initial wastewater and C_t_ is LVX content of the wastewater at the given time.

### 2.5. Experimental Design

The experimental design was executed by the Design Expert Software (version 8.0.6, Inc., Minneapolis, MN, USA). Five independent experimental variables, including current density, initial pH value, flow rate, chloride ion concentration, and reaction time, were controllable. These experiments were designed using the coded value and the central combination model. The experimental scheme is shown in Table 1.

### 2.6. Electrooxidation Experiment

Figure 2 shows the electrooxidation experimental device. The cyclic reaction mode was adopted. After the LVX wastewater was pumped into the electrochemical reactor, it was energized for the electrooxidation experiment. During the reaction, the aqueous solution in the electrochemical reactor was pumped back into the beaker through a pump. A rotor was placed in the beaker. The speed of the magnetic stirrer was adjusted to 300 r/min. After the reaction, the TOC was measured after filtration. To ensure that the reaction could proceed normally, 3% Na_2_SO_4_ was added to each group of reactions.

### 2.7. Model Fitting and Data Analysis

The quadratic equation model (Equation (3)) was applied to predict the response. Here, Y is the response; *b*_0_ is the offset term; *b_i_*, *b_ij_*_,_ and *b_ii_* are the linear, interaction effect, and squared effects, respectively; and ε is the random error.
(3)$$Y=b_{0}+\sum_{i=1}^{3}b_{i}x_{i}+\sum_{i=1}^{2}\sum_{i<j}^{3}b_{ij}x_{i}x_{j}+\sum_{i=1}^{3}b_{ii}x_{i}{}^{2}+\varepsilon$$

The data were analyzed using the analysis of variance (ANOVA) approach. The coefficient R^2^ expressed the quality of the fit of the polynomial model, whose statistical significance could be checked by the *F*-value. Then, the back elimination method was employed to optimize the model by selecting or eliminating the model terms according to the *p*-value with a 0.10 confidence level. Afterward, variables were chosen to estimate the optimized model’s accuracy. Finally, three-dimensional (3D) plots were obtained, and the interaction of the two factors on the responses was discussed.

### 2.8. Fluorescence Measurement

Hitachi’s F4700 was used to investigate the degradation process. For fluorescence studies, two measurement modes were employed. The first mode was employed to select the excitation wavelength at 286 nm and record the emission in the 250–600 nm range. The other mode was employed to select the emission wavelength at 510 nm and record the emission in the 250–600 nm range. The width of the excitation and emission slits was fixed at 5 nm.

## 3. Results and Discussion

### 3.1. Stability Assessment

#### 3.1.1. Scanning Electron Microscope (SEM) Analysis

As shown in Figure 3a,b, the scanning electron microscope was able to magnify by 1000 times; evidently, the surface of the titanium suboxide electrode is uneven and has a small amount of hole structure, and there is no obvious change before and after the reaction. Overall, the performance stability of the electrode was excellent. Figure 3a,b shows that the titanium suboxide electrode has a higher active surface area, which may promote the transfer of electrons and ions, thus increasing the specific capacitance in subsequent electrochemical detection [25,44]. To explore the distribution positions of O and Ti atoms in the titanium suboxide electrode, EDS detection on the titanium suboxide electrode was conducted. The green color in Appendix A represents the oxygen element. The oxygen element is uniformly distributed on the surface of the titanium suboxide electrode before and after the reaction, and even after the reaction, the oxygen element is more uniformly distributed. It is speculated that the electrooxidation process promotes the oxygen distribution of elements. In Appendix A, the red color represents titanium. Titanium is uniformly distributed on the surface of the titanium suboxide electrode before and after the reaction, indicating that the electrode has good stability.

According to Appendix A, the weight percentages of O and Ti before the reaction were 46.35% and 53.65%, and after the reaction they were 43.50% and 56.50%, respectively. The atomic percentages of O and Ti before the reaction were 72.12% and 27.88%, and after the reaction they were 69.74% and 30.26%, respectively, indicating that the electrode performance of stability was excellent.

#### 3.1.2. Atomic Force Microscopy (AFM) Analysis

As shown in Figure 3c,d, a 3D characterization of the apparent structure of the titanium suboxide electrode was performed. Some mountain-like structures were present on the surface of the titanium suboxide. Before the reaction, the average height was ~122.3 nm, and the average depth was about −58.4 nm. The electrode has a higher roughness and a larger specific surface area [44]. After repeated reactions, the average height and depth were ~116.4 and −55.7 nm, respectively. After repeated reactions, the edge of the electrode became unsharp, but the average height and depth did not change significantly, indicating that the electrode has good stability.

#### 3.1.3. X-ray Diffraction (XRD) Analysis

Appendix A shows that the surface crystal phase structure of the titanium suboxide electrode was analyzed using XRD, and the obvious Ti_4_O_7_ diffraction peaks appeared at diffraction angles of 14.3°, 31.7°, and 36.2°. Observing the XRD spectra of the titanium suboxide electrode before and after the reaction, almost no change occurred in the titanium suboxide electrode, which was basically consistent with the SEM results, indicating that the electrode prepared by the current method has good stability.

#### 3.1.4. XPS Analysis

From the results, Appendix A shows that titanium suboxide has two peaks at 464.32 eV and 458.49 eV, which belong to Ti2p1/2 and Ti2p3/2, respectively, indicating the existence of Ti^4+^. The characteristic peak at 457.90 eV belongs to Ti^3+^ [37,45].

### 3.2. Electrochemical Analysis

The electrochemical performance of the titanium suboxide electrode was investigated using the cyclic voltammetry of the titanium suboxide electrode (Appendix A). The titanium suboxide electrode has a wider electrochemical window, indicating high electrochemical activity. Appendix A shows the linear sweep voltammetry (LSV) curve of the titanium suboxide electrode. The LSV results show that the oxygen evolution potential of titanium suboxide (2.83 V vs. SCE) exceeded that of ruthenium–titanium (1.31 V vs. SCE), indicating that compared with the industrialized ruthenium–titanium electrode, the titanium suboxide electrode has higher electrochemical activity.

### 3.3. ANOVA and Model Simplification

Statistical analysis of the experimental data was performed to analyze the variance table (Table 2). Through the influence of this factor, the significance of the model simulated by the experimental design was verified. The model *F*-value of the TOC removal was less than unity, indicating that the noise variable of the model design is minute and the model has excellent significance [46]. Additionally, the lack-of-fit values of the model were small, indicating that the pure error effect during model design was minute. The results show that the model is significant and the best fitting effect is obtained.

As shown in Figure 4a, in the staggered normal distribution diagram, the residual points are evenly distributed on both sides of the fitted straight line, and most of the points fall on the straight line. Therefore, it could be inferred from Figure 4a that the fitted model is significant. Finally, Equation (3) was obtained.
(4)$$\begin{array}{cc} \mathrm{TOC}\;\mathrm{removal} & \;=\;0.29-7.496\times 10^{-3}A+0.067B-3.636\times 10^{-4}C+0.042D\\ & +\;6.842\times 10^{-3}E+2.689\times 10^{-3}\mathrm{AB}-5.705\times 10^{-4}\mathrm{AC}+7.075\times 10^{-3}\mathrm{AD}\\ & -\;0.026\mathrm{AE}-9.166\times 10^{-3}\mathrm{BC}-1.567\times 10^{-3}\mathrm{BD}-4.678\times 10^{-3}\mathrm{DE}\\ & -\;0.030A^{2}+8.603\times 10^{-3}B^{2}-0.014\mathrm{ABC}-0.015\mathrm{ABD}+0.015\mathrm{ADE}\\ & -\;0.042A^{2}B-0.035AB^{2} \end{array}$$
where A is the pH, B is the current density, C is the flow rate, D is the reaction time, and E is the chloride ion content.

### 3.4. Response Surface Single Factor Investigation

Figure 4b shows that as the current density in the reaction increases, the TOC removal gradually increases. The applied current density positively influences the degradation. It is assumed that when the current density was lower, the amount of hydroxyl radicals (·OH) produced was lower, which was insufficient to remove all pollutants. Figure 4c shows that as the circulating flow rate increases, the TOC removal changes insignificantly. Figure 4d shows that as the reaction time increases, the TOC removal gradually increases. As the reaction time increases, the electrode degrades the intermediate products, thus increasing the TOC removal. Figure 4e shows that as the chloride ion content increases, the TOC removal gradually increases. As the chloride ion content increases, active chlorine, such as HClO, will be generated in the water, which accelerates the mineralization of TOC. In Figure 4f, the TOC removal initially increases and then decreases as the pH increases. The reaction occurs best when the pH is neutral. Considering the cathode, it is speculated that a large amount of hydrogen peroxide could be produced in both acidic and alkaline conditions, and hydrogen peroxide could easily quench the ·OH generated on the anode, so the neutral conditions are better [47]. Considering the anode, it is speculated that in acidic solutions, the presence of HSO_4_^−^ may quench the ·OH, thereby reducing the degradation efficiency of LVX [48]. Alkaline conditions reduce the oxygen evolution overpotential and increase the side reaction of oxygen evolution. Therefore, the ion electrode shows lower removal efficiency under alkaline conditions [45]. It could be seen from single-factor investigation that pH and current density have a greater impact on TOC removal.

### 3.5. Response Surface Multifactor Interaction

From the 2D contour map in Figure 5, when the response variable was the TOC removal, the 2D contour map between the AB and AD factors was comprised of curves, indicating that the degree of influence of influencing factors on the response variable could be fully investigated within the range of value. The 2D contour map of AC, AE, BC, BD, and DE factors was a situation where curves and straight lines coexisted. This trend occurred because the range of value of the independent variables set in the experimental design is minute. There was an obvious interaction between A and E or A and D, and the analysis of the response surface diagram shows that a significant synergy exists between the AC and AE factors. However, the AE two-factor 3D response surface map was more prominent upwards. Thus, the AE two-factor antagonism was more obvious, and the best applicable value point was within the preset range of value. The A and B factors presented a concave shape toward the bottom surface, so the two factors antagonistically affected each other. Through the depression in the response surface, the lowest value of the fitted model could be determined, providing a reference range of value for the subsequent optimization process [46].

### 3.6. Experimental Verification

Based on the aforementioned experimental results and conditional analysis of factors, the optimal reaction conditions of the deep optimization factors are obtained through the simulation and optimization of the mathematical model: the current density, initial pH of the reaction, flow rate, chloride ion concentration, and reaction time were 39.6 A/m^2^, 6.5, 50 mL/min, 4%, and 120 min, respectively. Figure 6 shows the conversion of LVX during electrooxidation. Figure 6a shows that as time changes, the fluorescence intensity at the emission wavelength (510 nm) gradually decreases. Additionally, Figure 6b shows that the fluorescence intensity at the excitation wavelength (290 nm) decreased gradually with time. As shown in Figure 6, the 3D excitation-emission matrix (3D EEMs) fluorescence spectrum was employed in detecting the fluorescence change of the LVX solution during the electrooxidation process of the titanium suboxide electrode. As shown in Figure 6c, the three main peaks are E_x_/E_m_ = 250–300/450–600 (peak A), 300–375/450–600 (peak B), and 350–465/325–450 (peak C). According to Figure 6d, after 20 min of reaction, the fluorescence intensity of peaks A and B disappeared, whereas peak C increased, indicating that the conjugated heterocyclic structure of LVX was destroyed. As the reaction time increases, E_x_/E_m_ = 350–465/325–450 (peak C) in Figure 6g–i also disappear gradually, indicating that LVX was all converted into small molecules.

Figure 7a shows that under optimal reaction conditions, the removal rate of LVX reached 41%, which was basically consistent with the result predicted by the response surface (40.84%). As shown in Figure 7b, the LVX conversion rate of titanium suboxide reached 100%. The removal and conversion rates of LVX are significantly greater than that of the ruthenium–titanium electrode, indicating that the titanium suboxide electrode has much better electrochemical performance than the industrially produced ruthenium–titanium electrode. For the EPR test, spin trapping was employed on 5,5-dimethyl-1-pyrroline-1-oxide (DMPO) as a hydroxyl-radical scavenger (Figure 7e). As the reaction progresses, the intensity of hydroxyl radicals gradually increases.

### 3.7. Exploration of Degradation Mechanism

Tert-butyl alcohol (TBA) has been frequently adopted as a quenching agent for hydroxyl radicals (·OH). Hence, herein, TBA was added to the system under study to elucidate the degradation mechanism. In this reaction, active chlorine mainly stems from NaCl added in the reaction. Figure 7c shows that after adding TBA, the TOC removal and the LVX conversion rate decreased, indicating that ·OH influenced the degradation of LVX. Active chlorine has a promoting effect on the conversion and mineralization of LVX when NaCl is not involved in the reaction (Equations (4)–(6)) [16]. Figure 7d shows that when NaCl was absent, the removal rate of TOC and the conversion rate of LVX decreased more, so it was speculated the active chlorine significantly influences the degradation of LVX, followed by ·OH.
2Cl^−^ → Cl_2_ + 2e^−^(5)
Cl_2_ + H_2_O ⇄ HClO + Cl^−^ + H^+^(6)
HClO ⇄ ClO^−^ + H^+^.(7)

### 3.8. Possible Degradation Routes of LVX

To further disclose the degradation mechanism, the intermediate products of LVX degradation in the electrooxidation system were identified using LC–MS. Figure 8 displays the MS spectra of detected degradation intermediates. Furthermore, five plausible degradation routes of LVX were proposed (Figure 8) according to the intermediates. In pathway I, the hydroxylation reactions lead to the production of L1 (*m*/*z* = 379). L6 (*m*/*z* = 337) was formed by decarboxylation of L1 (*m*/*z* = 379). In pathway III, according to this, the molecular ion peak underwent a decarboxylation reaction of the methyl morpholine group in the LVX drug and was transformed into L3 (*m*/*z* = 333). Then, the decarboxylation and despiperazine groups of L3 (*m*/*z* = 333) lead to the production of L10 (*m*/*z* = 250). In pathway IV, L4 (*m*/*z* = 278) was initially formed via an attack on the N-methyl piperazine group by reactive radicals (·OH) and active chlorine. L4 (*m*/*z* = 278) was converted to L10 (*m*/*z* = 250) by decarboxylation. In pathway V, L5 (*m*/*z* = 317) was produced via the decarboxylation of LVX. Furthermore, L14 (*m*/*z* = 163) was obtained by demethylation, decarboxylation, and despiperazine groups of L5. In pathway II, the demethylation and hydroxylation reactions lead to the production of L2 (*m*/*z* = 363). L7 (*m*/*z* = 335) was formed by the decarboxylation of L2 (*m*/*z* = 363). In addition, L6 (*m*/*z* = 337) could be obtained by breaking the double bond of L7 (*m*/*z* = 335). L7 (*m*/*z* = 335), the important intermediate of LVX, was further degraded on the N-methyl piperazine ring, and it was oxidized to form a stable intermediate form L8 (*m*/*z* = 264). Afterward, the intermediate compound L8 was further dealkylated to produce two intermediates: L9 (*m*/*z* = 250) and L10 (*m*/*z* = 250). Next, the demethylation and dehydroxylation lead to the production of L11 (*m*/*z* = 234) and L12 (*m*/*z* = 234). Furthermore, the intermediate compounds L11 and L12 were further dealkylated to produce two intermediates, L13 (*m*/*z* = 181) and L14 (*m*/*z* = 163), via decarboxylation, the opening of quinolone, and demethylation. L13 and L14 compounds could further undergo defluorination and hydroxylation to form L15 (*m*/*z* = 93). Subsequently, the opening of benzene rings lead to the production of L16 (*m*/*z* = 80) and L17 (*m*/*z* = 54). Finally, intermediate L17 products might be continuously broken into small-molecule organic acids and mineralization products [49,50,51,52,53,54,55,56].

### 3.9. Treatment of Wastewater from Novel Coronavirus Epidemic

Titanium suboxide electrodes were employed to treat wastewater from the novel coronavirus pneumonia epidemic. Figure 9a shows the results. The removal rate of sulfamethoxazole, azithromycin, and LVX reached 100%, and the total removal rate of all antibiotics reached 94.5%, indicating that the titanium suboxide electrode has great development potential. Figure 9b,c show the industrial wastewater treatment device.

## 4. Conclusions

Herein, bench-scale electrooxidation equipment for hospital wastewater was used to evaluate the removal effect of the primary antibiotic, levofloxacin (LVX), in hospital wastewater, which proved the stability and high efficiency of the equipment. In this bench-scale apparatus, titanium suboxide and ruthenium–titanium were used as the anode and cathode, respectively. RSM was used to conduct electrooxidation and degradation of LVX model wastewater. In the degradation process, the influence of current density, initial pH value, flow rate, chloride ion concentration, and reaction time on the TOC removal were investigated and the best reaction condition was obtained as follows: current density, initial aqueous pH, flow rate, concentration of chloride ion, and reaction time were 39.6 A/m^2^, 6.5, 50 mL/min, 4% and 120 min, respectively. The TOC removal could reach 41% after reaction time 120 min, which was basically consistent with the result predicted by the response surface (40.84%).

The morphology and properties of the electrode were analyzed using techniques such as SEM, XRD, EDS, and AFM, which showed that the titanium suboxide electrode has high stability and is a promising electrode material.

Further, the bench-scale electrooxidation equipment was changed into onboard-scale electrooxidation equipment, and its application was promoted in several hospitals in Dalian. The results show that the removal rate of titanium suboxide for sulfamethoxazole, azithromycin, and LVX reached 100%, and the total removal rate of all antibiotics reached 94.5%, indicating that the titanium suboxide electrode has great development potential.

## Figures and Tables

**Figure 1 materials-14-06814-f001:**
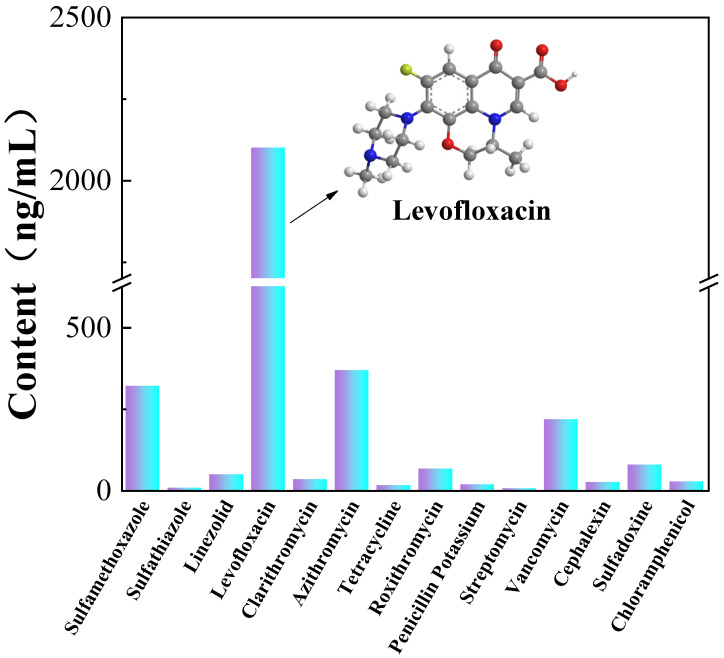
Antibiotic content in wastewater from novel coronavirus pneumonia epidemic.

**Figure 2 materials-14-06814-f002:**
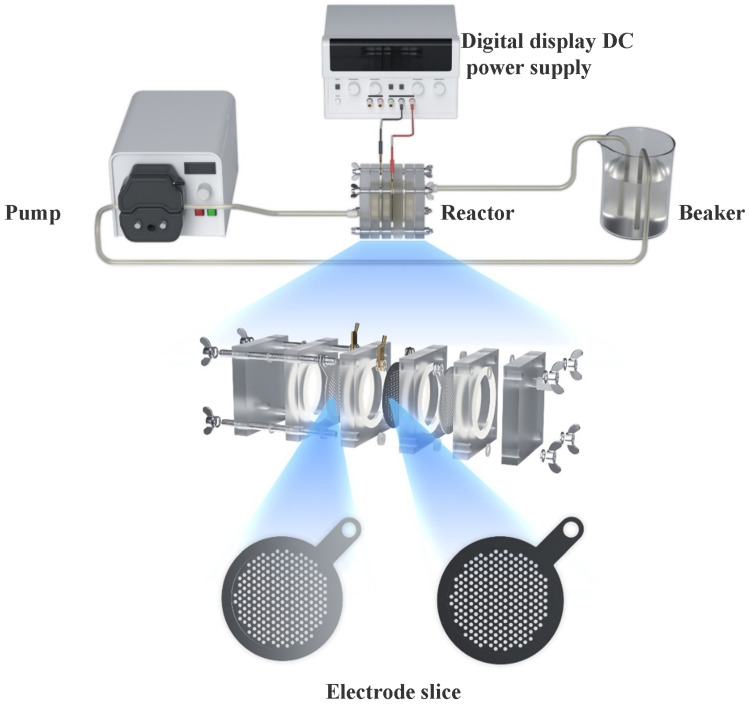
Electrooxidation experimental device.

**Figure 3 materials-14-06814-f003:**
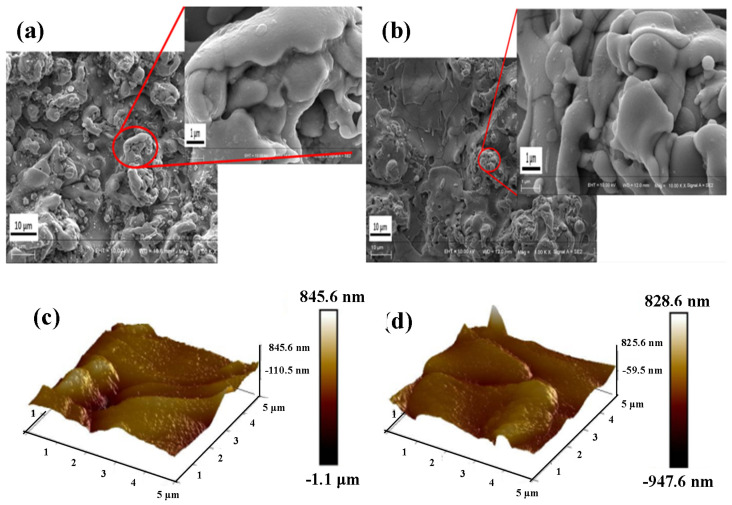
SEM spectrum before (**a**) and after (**b**) electrooxidation reaction. AFM before (**c**) and after (**d**) electrooxidation reaction.

**Figure 4 materials-14-06814-f004:**
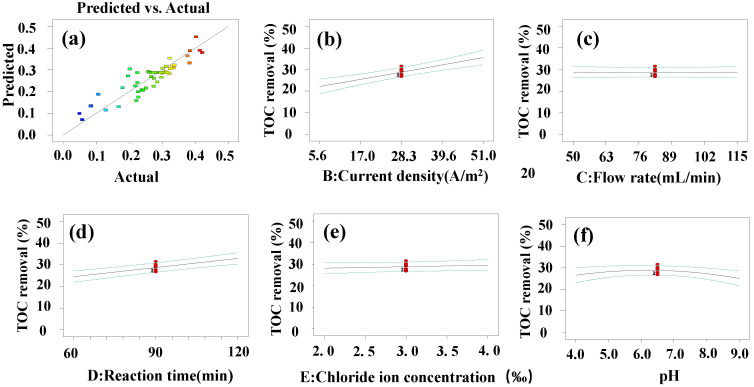
(**a**) Staggered normal distribution diagram. (**b**–**f**) Influence of single factor on TOC removal.

**Figure 5 materials-14-06814-f005:**
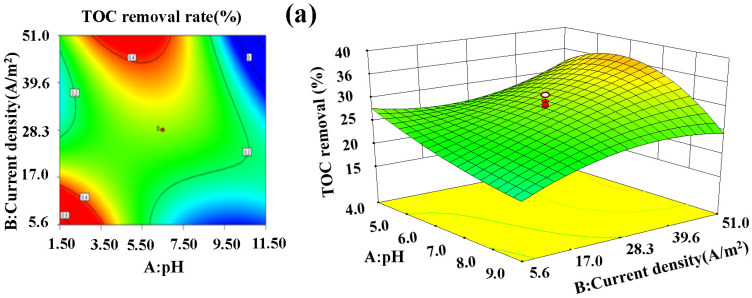
Two-dimensional contour maps (**left**) and three-dimensional response surface diagrams (**right**) of the TOC removal of: (**a**) factors A, B (flow rate, chloride ion concentration, and reaction time of 82.5 mL/min, 3‰, and 90 min, respectively); (**b**) factors A, C (current density, chloride ion concentration, and reaction time of 28.3 A/m^2^, 4‰, and 120 min, respectively); (**c**) factors B, C (initial pH, chloride ion concentration, and reaction time of 6.5, 90 min, and 3‰, respectively); (**d**) factors B, D (initial pH, flow rate, and chloride ion concentration of 6.5, 82.5 mL/min, and 3‰, respectively); (**e**) factors E, D (current density, initial pH, and flow rate of 28.3 A/m^2^, 6.5, and 82.5 mL/min, respectively); (**f**) factors A, D (current density, flow rate, and chloride ion concentration of 28.3 A/m^2^, 82.5 mL/min, and 3‰, respectively); and (**g**) factors A, E (current density, flow rate, and reaction time of 28.3 A/m^2^, 82.5 mL/min, and 90 min, respectively).

**Figure 6 materials-14-06814-f006:**
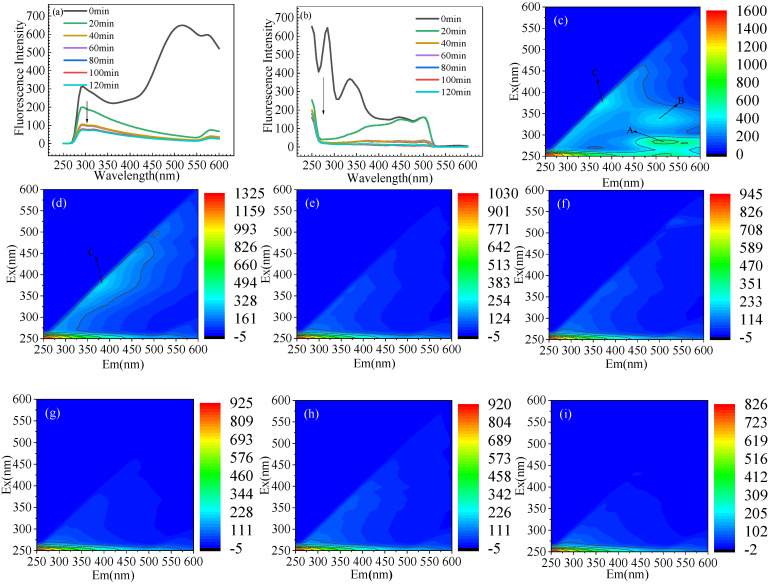
(**a**) Fluorescence spectra of LVX at emission wavelength of 510 nm. (**b**) Fluorescence spectra of LVX at 290 nm excitation wavelength. Three-dimensional EEMs of LVX solution after electrocatalysis degradation of (**c**) 0, (**d**) 20, (**e**) 40, (**f**) 60, (**g**) 80, (**h**) 100, and (**i**) 120 min by titanium suboxide anode.

**Figure 7 materials-14-06814-f007:**
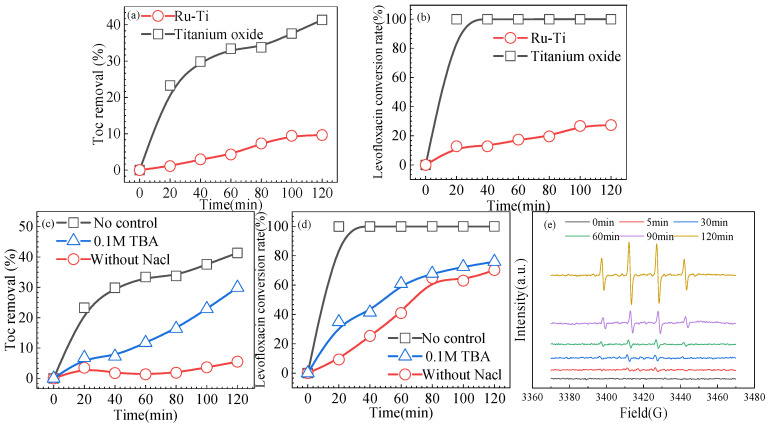
(**a**) TOC removal and (**b**) LVX conversion rates of ruthenium–titanium electrode and titanium suboxide under optimal reaction conditions (current density of 39.6 A/m^2^, initial pH of 4, flow rate of 50 mL/min, chloride ion concentration of 4%, and reaction time of 120 min). (**c**) Effect of NaCl and TBA on TOC removal. (**d**) Effect of NaCl and TBA on LVX conversion. (**e**) EPR spectra of hydroxyl radicals.

**Figure 8 materials-14-06814-f008:**
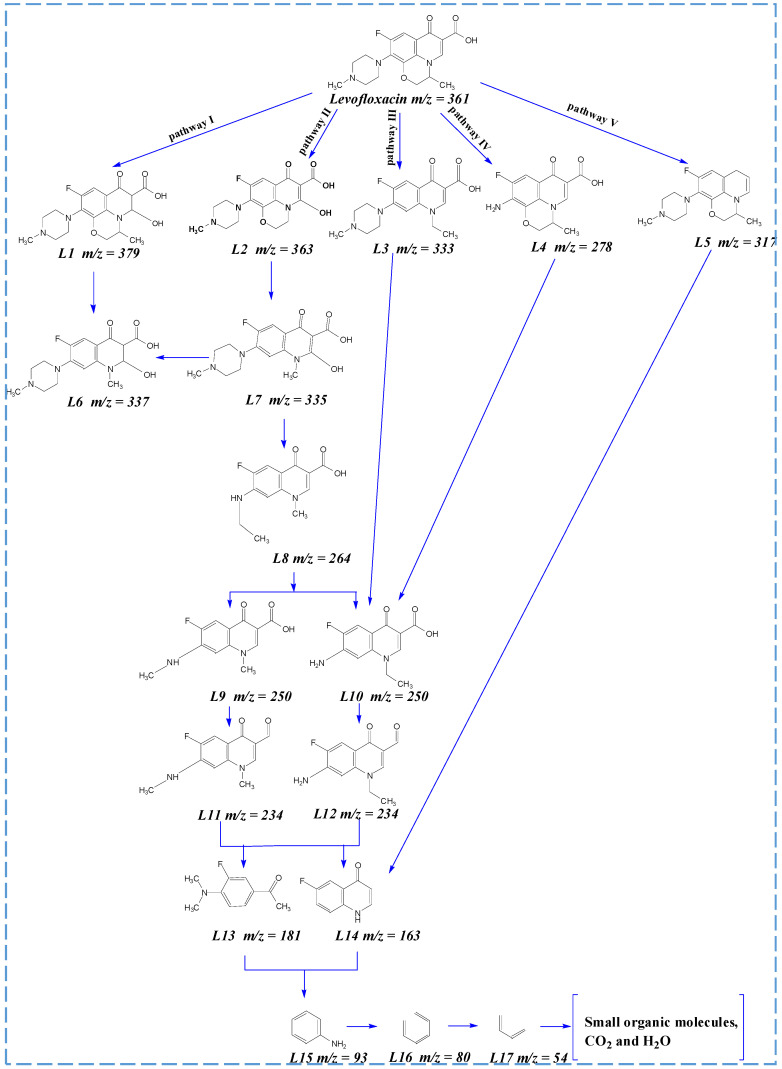
Proposed degradation pathways of LVX during electrooxidation reaction.

**Figure 9 materials-14-06814-f009:**
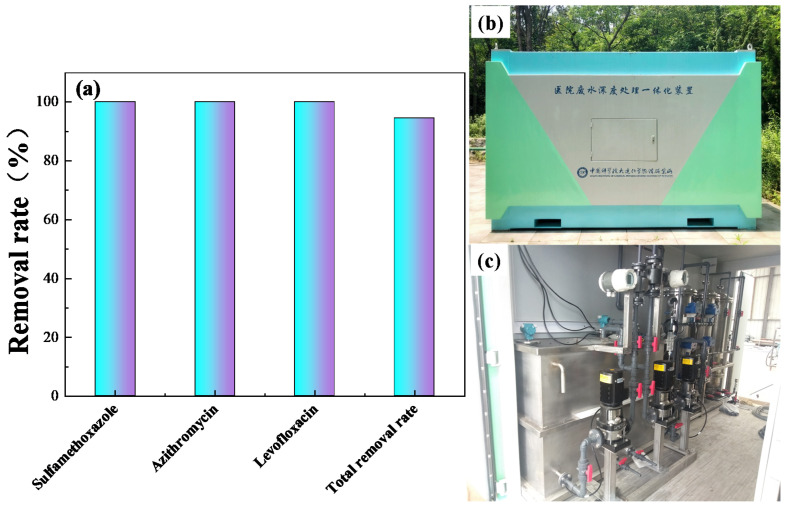
(**a**) Treatment of wastewater from novel coronavirus epidemic. (**b**) and (**c**) Integrated device for advanced treatment of hospital wastewater.

**Table 1 materials-14-06814-t001:** Actual value and code value correspondence table.

Factor Name	Number	Unit			Coded Value		
			−1.68	−1	0	1	1.68
Initial pH value	A		0.55	4	6.5	9	12.4
Current density	B	A/m^2^	5.6	17	28.3	39.6	51.0
Flow rate	C	mL/min	5.20	50	82.5	115	159.7
Reaction time	D	min	18.6	60	90	120	161.3
Chloride ion content	E	‰	0.62	2	3	4	5.38

**Table 2 materials-14-06814-t002:** ANOVA.

Source	Sum of Squares	df	Mean Square	*F*-Value	*p*-Value Prob > *F*	Significance
Model	3.10 × 10^−1^	19	1.63 × 10^−2^	6.95	<0.0001	significant
A-pH	6.36 × 10^−4^	1	6.36 × 10^−4^	2.71 × 10^−1^	0.6067	-
B-current density	5.11 × 10^−2^	1	5.11 × 10^−2^	21.76	<0.0001	-
C-flow rate	5.73 × 10^−6^	1	5.73 × 10^−6^	2.44 × 10^−3^	0.9609	-
D-reaction time	7.72 × 10^−2^	1	7.72 × 10^−2^	32.89	<0.0001	-
E-chloride ion concentration	2.03 × 10^−3^	1	2.03 × 10^−3^	8.64 × 10^−1^	0.3601	-
AB	2.31 × 10^−4^	1	2.31 × 10^−4^	9.85 × 10^−2^	0.7558	-
AC	1.04 × 10^−5^	1	1.04 × 10^−5^	4.43 × 10^−3^	0.9473	-
AD	1.60 × 10^−3^	1	1.60 × 10^−3^	6.82 × 10^−1^	0.4154	-
AE	2.17 × 10^−2^	1	2.17 × 10^−2^	9.25	0.0049	-
BC	2.69 × 10^−3^	1	2.69 × 10^−3^	1.14	0.2932	-
BD	7.86 × 10^−5^	1	7.86 × 10^−5^	3.35 × 10^−2^	0.8561	-
DE	7.00 × 10^−4^	1	7.00 × 10^−4^	2.98 × 10^−1^	0.5890	-
A^2^	5.31 × 10^−2^	1	5.31 × 10^−2^	22.63	<0.0001	-
B^2^	4.29 × 10^−5^	1	4.29 × 10^−5^	1.83 × 10^−2^	0.8934	-
ABC	6.21 × 10^−3^	1	6.21 × 10^−3^	2.64	0.1145	-
ABD	7.57 × 10^−3^	1	7.57 × 10^−3^	3.22	0.0827	-
ADE	7.63 × 10^−3^	1	7.63 × 10^−3^	3.25	0.0816	-
A^2^B	1.48 × 10^−2^	1	1.48 × 10^−2^	6.30	0.0177	-
AB^2^	1.00 × 10^−2^	1	1.00 × 10^−2^	4.26	0.0478	-
Residual	7.04 × 10^−2^	30	2.35 × 10^−3^	-	-	-
Lack of Fit	6.83 × 10^−2^	23	2.97 × 10^−3^	9.77	0.0024	significant
Pure error	2.13 × 10^−3^	7	3.04 × 10^−4^	-	-	-
Cor total	0.38	49	-	-	-	-

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
