# Peer review of "Electrocatalytic Degradation of Levofloxacin, a Typical Antibiotic in Hospital Wastewater"

_materials, 2021, doi:10.3390/ma14226814_

Round 1

Reviewer 1 Report

Authors reported a potential method for treatment of antibiotic levofloxacin in hospital wastewater from novel coronavirus epidemic by electrocatalysis. Authors developed active and stable titanium suboxide electrode which demonstrated as a suitable candidate in this application. Authors used potential method of response surface methodology (RSM) for optimizing the reaction followed by experimental validation. I think that this is of great interest responding to current trend for pollutant removal technologies in this pandemic. However, I cannot accept this manuscript in the current form for publishing in this journal. Authors should edit according to below comments:

a) Language and writing: Authors much carefully proofreading of manuscript again. There are numbers of language problems.

  1. Title contains two “epidemic” words. Please double check if it is intentionally used so, or problem?
  2. In the abstract: authors suddenly use the short cut of LVX. They should mention in the early sentence what are full words of LVX?
  3. Authors should unify the writing style in a whole manuscript. For example, in Page 4, Paragraph 1, [Jiang et employ ….] this sentence is written in present tense. But the rest sentences in that paragraph is in past tense. Please unify it and check a whole manuscript.

b) Scientific comments:

  1. Figure 2: authors should explain what are of the parts (components)? For example, beaker, pump …? Readers might not recognize what they are without any notation by text.
  2. Page 9, authors mentioned “Fig. 3 (a) and (b) shows that this titanium suboxide electrode has a higher active surface area, which may promote the transfer of electrons and ions, so it may provide more specific capacitance in subsequent electrochemical detection [23, 41].” How do they know high surface area with SEM image?
  3. In Figure 3 shows images of before and after reaction. Authors should explain which reaction time? There are many reaction times in the manuscript.
  4. For me, preparation of electrode is one of the key roles in this work. Electrode characterizations are important. I recommend authors to use XPS or EPR/ESR for analysis of oxidation state of titanium (e.g. Ti4+, Ti3+..) because samples were reduced under H2.
  5. Too many subtitles with short explanation. For example, 3.4.1~3.4.5 subtitles contain very short explanation. Some subtitles contain only one sentence. Authors should combine them together or try to explain more of each subtitle.
  6. Figure 5’s caption should explain what is (a), (b)….
  7. Figure 6: label of figure of (c) to (i) should be in white color. It is easy to see.
  8. In 3.4.5 Initial pH optimization, authors mention pH neutral was the best performance. But, in experimental verification authors used pH 4. Please explain the reason.
  9. MS spectra should be included in supporting information.
  10. There is no detail information related to experiments of on-board-scale electro-oxidation (e.g. reaction conditions).
  11. Figure 9(b) and (c) show the integrated device for advanced treatment of hospital wastewater. Where is device from? Did they construct this system?
  12. I cannot find Supplementary Information attached to manuscript.

Author Response

Thank you for your letter and for the reviewers’ comments concerning our manuscript entitled “Electrocatalytic degradation of levofloxacin, a typical antibiotic in hospital wastewater from the novel coronavirus epidemic (ID: materials-1386639). Those comments are all valuable and very helpful for revising and improving our paper, as well as the important guiding significance to our researches. We have studied comments carefully and have made correction which we hope meet with approval. 

Reviewer 2 Report

English of the paper is not on the required level. This which makes the reading of the manuscript difficult.

An introduction includes unnecessary details of published works on application of RSM approach. The description of the own work is too long.  There is no reference for data in Fig. 1.

The full list of routine experimental instruments is not necessary to provide.

I did not understand the electrode preparation section due to bad English.

Eq.1, TOCt is not “the total organic carbon of the levofloxacin (LVX) wastewater,” but TOC of all organic material including oxidation products of LVX. “TOC removal” is the unitless value, not the rate

The electrooxidation experiment description is missing the important information. What is the “cyclic reaction mode?” “The TOC was measured after filtration.” What is the precipitation? Was the initial solution homogeneous?  What was the supporting electrolyte, 3‰ Na2SO4?

The content of the section 2.8 is not consistent with the title of the section.

Section 3.1.1. “there is no obvious change before and after the reaction.” The reaction conditions and the reaction time are not described. Therefore, the statement “Overall, the electrode performance of stability was excellent” is meaningless. The same is applicable for the sections 3.1.2 and 3.1.3.

Section 3.2. Figs S2a-S2B does not provide any information on electrochemical activity

Eq. 3 is for the “TOC removal rate.” How has it been measured? What are the units of this value? Eq. 1 is for the “TOC removal.”

Section 3.3 is the manipulations with numbers, which is difficult to understand.

Fig. 7. “TOC removal rate” and “Conversion rates” are not defined and are unitless (%)

The reactions in eqs 5-6 take place under basic conditions.

Section 3.8 is highly speculative and is not confirmed with experimental data. If Cl2 is involved, it must result in formation of chlorinated products. If the reaction is carried out under air, than O2 should scavenge all carbon centered radicals.

General concerns.

The experimental part is leaking the important information. What actually has been measured, TOC or LVX concentration? TOC might vary in a large range at the same LVX concentration. What 100% TOC means? The full mineralization LVX to CO2? Is it necessary to achieve 100% TOC? What if acetic acid is an intermediate product? This is a biodegradable product, which is extremely difficult to oxidize.

The electrochemical studies are not on the minimally required level. In the reactor used in this study the cathode and anode are not separated spatially, the cross reactions are inevitable. The applied potentials are not specified, the cell should have a very high ohmic resistance. The applied potential must be included as the major factor in analysis. The Faradaic efficiency is not estimated and is likely to be unacceptably low.  

Author Response

(The authors gave the same response as above.)

Reviewer 3 Report

Why in the title is 'novel coronavirus epidemic epidemic".  It has nothing to do with experiments.  Seems like authors want to have some catching words.

Has levofloxacin some special function in coronavirus treatment?

Some minor points to correct;

1 in Abstract "and the reaction time was 120 min, the removal rate of levofloxacin could reach 41%,"  should be changed as reaction time was arbitrary chosen. "the removal of levofloxacin could reach 41% after reaction time 120 min"

2 on Figure 5 a, bc, d, e, f, g  there are 3 D mesh visualization of two out of more variables. In my opinion  for every of visualization values of constant other varables should be shown in a, b, c etc captions.  For Fig. 5 a which presents dependence of current density and pH on TOC removal, flow rate , reaction time and chloride ion concentration values should be shown.

3 Seems that flowrate has no influence on TOC, so it should be excluded from equation. Other point is optimum flow rate - if it exists - is characteristic for this exact experimental setup.  If this flowrate has meaning then other parameters like mean residence time and geometrical sizes of electrodes should be specified. It is obvious when this treatment equipment will be built for other volumes of hospital wastewater.

Author Response

(The authors gave the same response as above.)

Round 2

Reviewer 1 Report

Manuscript has been significantly improved

Author Response

Thank you for your letter and for the reviewers’ comments concerning our manuscript entitled “Electrocatalytic degradation of levofloxacin, a typical antibiotic in hospital wastewater (ID: materials-1386639). Those comments are all valuable and very helpful for revising and improving our paper, as well as the important guiding significance to our researches. We have studied comments carefully and have made correction which we hope meet with approval. Here, we attached revised manuscript in the formats of both PDF and word. A document answering every question from the referees was also summarized and enclosed. Here below is our description on revision according to the reviewers’ comments.

Reviewer 2 Report

The minor revisions of the original version did not improve the manuscript. We authors did not address my key questions. English is still far from minimally required. The electrochemical part does not meet even the most modest standards. Below are several examples.

  1. The design of electrochemical cell is not described. The shape of electrodes in Figure 2 are not consistent with the description of the anode preparation. The reactions on cathode are not even mentioned, the direction of solution flows (through cathode and then anode or vice versa) is not specified. The cross reactions in such systems are inevitable. No information on applied potentials, ohmic resistance, amount of charged transferred, Faradaic yield, etc. Based on my best understanding, the two electrodes were immersed into flowing solution and some potential was applied to electrodes.
  2. The terms “removal rate” and “conversion rate” are not used in kinetics. Units of rates always include time. Eqs 1-2 are the definition of conversion, which is time dependent. Numerous statements are intrinsically contradictive. The example is the caption to the Figure 7. TOC removal and LVX conversion are plotted as a function of time, while in the caption is written “reaction time 120 min.”
  3. In the Author’s response the reactions in eqs 5-6 are reversible, but not in the main text. The formation of HClO at pH 4 or 6.5 is very questionable. TBA is not good HO-radical scavenger. Chlorine means Cl2, but what “the active chlorine” is not specified.

I could add more questions, but the 3 points listed above are sufficient to reject the paper.

Author Response

(The authors gave the same response as above.)

Reviewer 3 Report

Some minor points

1) For Figure 5, in my opinion it would be better to put "(flow rate 82.5 mL/min, chloride ion concentration 3‰, reaction time 90 min," just below a) diagram and the same for other ie. b, c, d, e, f, g.

2) "indicating that the titanium suboxide electrode has excellent electrochemical performance than the industrially produced ruthenium–titanium electrode"  change excellent to much better

3) In the coverletter there is: "According to the literature[12], Levofloxacin (LVX) is the most widely used one, mainly for the treatment of pneumonia, urinary tract infection, acute pyelonephritis, skin and tissue infections.", however [12] is not about levofloxacin treatment. And this reference is not 2022, but rather Chemical Engineering Journal 428, 131257

4) I am not convinced that in the title of the paper "coronavirus" is neccessary. According to many sources "Levofloxacin Oral: Uses, Side Effects, Interactions ... - WebMD

https://www.webmd.com
 Uses. Levofloxacin is used to treat a variety of bacterial infections. ... It will not work for viral infections (such as common cold, flu)."

Author Response

(The authors gave the same response as above.)
